# The Impact of Environmental Sustainability Labels on Willingness-to-Pay for Foods: A Systematic Review and Meta-Analysis of Discrete Choice Experiments

**DOI:** 10.3390/nu13082677

**Published:** 2021-07-31

**Authors:** Anastasios Bastounis, John Buckell, Jamie Hartmann-Boyce, Brian Cook, Sarah King, Christina Potter, Filippo Bianchi, Mike Rayner, Susan A. Jebb

**Affiliations:** 1Nuffield Department of Primary Care Health Sciences, University of Oxford, Radcliffe Observatory Quarter, Woodstock Road, Oxford OX2 6GG, UK; Anastasios.Bastounis1@nottingham.ac.uk (A.B.); john.buckell@ndph.ox.ac.uk (J.B.); Jamie.hartmann-boyce@phc.ox.ac.uk (J.H.-B.); seking100@googlemail.com (S.K.); Christina.Potter@Nicklaushealth.org (C.P.); filippo.bianchi@phc.ox.ac.uk (F.B.); susan.jebb@phc.ox.ac.uk (S.A.J.); 2Division of Epidemiology & Public Health, School of Medicine, University of Nottingham, City Hospital, Hucknall Road, Nottingham NG5 1PB, UK; 3Nuffield Department of Population Health, University of Oxford Richard Doll Building, Old Road Campus, Oxford OX3 7LF, UK; mike.rayner@ndp.ox.ac.uk

**Keywords:** willingness to pay, sustainable, organic, meta-analysis, ecolabels

## Abstract

Food production is a major contributor to environmental damage. More environmentally sustainable foods could incur higher costs for consumers. In this review, we explore whether consumers are willing to pay (WTP) more for foods with environmental sustainability labels (‘ecolabels’). Six electronic databases were searched for experiments on consumers’ willingness to pay for ecolabelled food. Monetary values were converted to Purchasing Power Parity dollars and adjusted for country-specific inflation. Studies were meta-analysed and effect sizes with confidence intervals were calculated for the whole sample and for pre-specified subgroups defined as meat-dairy, seafood, and fruits-vegetables-nuts. Meta-regressions tested the role of label attributes and demographic characteristics on participants’ WTP. Forty-three discrete choice experiments (DCEs) with 41,777 participants were eligible for inclusion. Thirty-five DCEs (n = 35,725) had usable data for the meta-analysis. Participants were willing to pay a premium of 3.79 PPP$/kg (95%CI 2.7, 4.89, *p* ≤ 0.001) for ecolabelled foods. WTP was higher for organic labels compared to other labels. Women and people with lower levels of education expressed higher WTP. Ecolabels may increase consumers’ willingness to pay more for environmentally sustainable products and could be part of a strategy to encourage a transition to more sustainable diets.

## 1. Introduction

Food production systems are one of the main contributors to environmental degradation across the globe [1]. The United Nations Sustainable Development Goals framework calls for governments to foster more sustainable food production systems [2,3]. However, changing food production systems to enhance environmental sustainability may impose costs which, in turn, may be passed on to consumers [4]. Price is a significant determinant of consumer behaviour [5]. Higher prices reduce demand [6] and price reductions are a common strategy to increase sales [7]. It follows that higher prices for environmentally sustainable products may deter consumers, which could be a barrier in the transition to more sustainable food systems. 

One way to potentially mitigate the impact of higher prices when consumers are choosing between more and less environmentally sustainable products would be to increase the appeal of the more sustainable option. One way of doing this is through the use of environmental impact labels on products (hereafter, ecolabels), to highlight more (or less) sustainable products [8,9]. Providing consumers with accurate and detailed information about environmental impacts can help raise their awareness and support more informed product choices. The addition of labels may also prompt businesses to reduce the environmental impacts of their products to avoid potentially negative public attention and reduced sales. 

In a companion review, we found that ecolabels can increase the selection and purchase of more environmentally sustainable food and drink products, regardless of message type (e.g., organic or “environmentally friendly”) and format (e.g., text or logo) [10]. Though some studies have tested consumers’ willingness to pay (WTP) for ecolabelled products [11], there is a lack of consolidated data on how exposure to different types of ecolabels affects the perceived value of a product, expressed in monetary terms, and whether this varies with demographic variables such as age, gender, income, or for different types of products.

The aims of this review are: (i) to systematically review and quantitatively synthesise the experimental evidence regarding the prices that consumers are willing to pay for foods, including milk, with an ecolabel compared to products without ecolabels; (ii) to explore whether consumers’ willingness to pay for labelled foods/milk changes as a result of different messages/ecolabel attributes (e.g., certification); and, (iii) to investigate the associations between demographic variables and WTP. 

## 2. Materials and Methods

We conducted a systematic review and meta-analysis of discrete choice experiments. Reporting follows the preferred reporting items for systematic reviews and meta-analyses (PRISMA) checklist [12]. The protocol was pre-registered with PROSPERO (CRD42018094330). The methods for searching, screening, and data extraction followed those described in the Cochrane Handbook for Systematic Reviews of Interventions [13]. 

### 2.1. Eligibility Criteria and Search Strategy

The search strategy was developed alongside a companion review in collaboration with an experienced librarian (N. R.) [10]. Initially, we searched EMBASE, MEDLINE, PsycINFO, Social Science Citation Index (SSCI), CAB Abstracts, the Cochrane Controlled Register of Trials (CENTRAL), and the Cochrane Database of Systematic Reviews from inception to 20 April 2019 using a combination of Medical Subject Headings (MeSH terms) and free-word terms in three key areas: (i) intervention characteristics (e.g., ecolabels), (ii) conditions of interest (e.g., eating preferences, diets), (iii) outcomes (e.g., “willingness to pay”, purchase), and (iv) method of analysis (e.g., discrete choice modelling, conjoint-based analysis). The detailed search strategy is available in Appendix B. An updated, more precise, follow-up search adding in search filters specific to studies measuring willingness to pay was conducted on 8 November 2019 using the same databases. The search filters consisted of the following terms: ‘discrete choice experiment,’ ‘discrete choice experiments’, ‘discrete choice modelling’, ‘discrete choice conjoint experiment’, ‘discrete-choice experiment’, ‘discrete-choice experiments’, conjoint analysis’, ‘conjoint measurement’, ‘conjoint studies’, ‘conjoint choice experiment’, ‘conjoint choice experiments’, ‘Best-Worst Scaling’, ‘Best Worst Scaling’, ‘MaxDiff Scaling’, ‘Maximum Difference Scaling’, ‘Contingent valuation’. 

We included studies that provided measures of consumers’ WTP through in-person interviews/questionnaires in two settings, stores/supermarkets and virtual environments (e.g., online experimental platforms). We included DCEs that evaluated the effects of ecolabels on consumers’ WTP for foods, including milk, which compared ecolabels to no label or to other labels unrelated to the environment e.g., health labels. Eligible ecolabels were classified as: (i) labels conveying messages relevant to organic production of foods and milk, or (ii) labels conveying messages relevant to environmental sustainability attributes of foods and milk (such as carbon dioxide emissions, water efficiency, land use, pesticide use, and the impact of production methods on biodiversity). All formats of ecolabels (i.e., text, logo, combined) were eligible for inclusion. For studies in which there were multiple intervention arms, only data comparing the intervention to the control (no ecolabel) arms were analysed. No restrictions on population or geography were imposed. Only studies published in English were eligible for inclusion in this review. The outcome of interest was the monetary value that respondents place on the presence of the ecolabel on a product (i.e., how much people are willing to pay for it) over and above a product with no such labelling, expressed as the marginal WTP (MWTP) [14]. 

We contacted authors for further information when full-text articles were not available, there was insufficient information provided for the inclusion criteria to be applied, or there were insufficient details reported on the outcomes. Authors were contacted twice before a decision was made. In total, 17 authors were contacted and we received replies from 12 (71% response rate). Lack of reply from some authors or inability of some authors to provide the requested information led us to exclude eight studies from the meta-analysis; these are reported narratively [15,16,17,18,19,20,21,22].

### 2.2. Data Extraction and Quality Assessment

Following the initial screening of titles and abstracts, full-texts of all potentially relevant studies were assessed for inclusion independently by two reviewers. Disagreements were resolved through discussion or referral to a third reviewer. Two independent reviewers (AB and SK/CP/FB) conducted full-text screening with a high-level of agreement (κ = 0.96). Two reviewers extracted data independently and disagreements were resolved through discussion (level of agreement in extraction phase > 85%). A predefined, pilot-tested data extraction form was used, including information on study authors and funding sources, consumers’ demographic characteristics, setting (in-person or online), intervention and comparator characteristics, the method for assessing the outcome type of products, characteristics of the DCEs analysis (e.g., DCE design, estimation procedures, and validation tests), numerical outcome data (e.g., MWTP in monetary value per unit [e.g., per kg], plus any available measure of variability or percentage of difference in MWTP where this was available) at any available assessment point. Two study authors independently assessed the quality of included DCEs, using the International Society for Pharmacoeconomics and Outcomes Research (ISPOR) checklist [23,24]. The assessment of the quality of reporting of the included DCEs targeted the following categories: (i) study perspective and research question, (ii) choice of attributes and levels, (iii) construction of tasks, (iv) choice of experimental design, (v) method for eliciting consumers’ preferences, (vi) data collection instrument, (vii) data collection plan, (viii) statistical analyses and model estimation, (ix) validity of results, and (x) quality in reporting the findings.

As most studies reported demographic characteristics using categories (e.g., 32% of participants were <35 years of age, 68% were >35 years of age) rather than means (e.g., the average age of participants was 52.3 years), we treated demographic variables as categorical predictors (e.g., percentage within specific ranges) in our meta-analyses. We extracted and analysed demographic data based on ranges to use as much of the data as possible. Studies that reported means or medians of consumers’ demographic characteristics were not included in the analysis of demographic factors as the form of the data would not allow for this. 

### 2.3. Data Analysis

Country-specific mean MWTP for foods were transformed to purchasing power parity dollars (PPP$), using the Organization for Economic Cooperation and Development (OECD) database [25]. Based on the World Bank database [26], the PPP$ values were adjusted to country-specific inflation rates (ex post), using 2018 as the base year. When the year was not reported, we proxied the year using the year that was two years before the study was published. Although the ex post estimation is not the only method for adjusting WTP values to inflation rates [27], it has been suggested as the preferred method [28] and has been used previously [29]. All effect sizes imported in the analysis were standardised to a single measurement unit (per kilogram), as mass was the most reported measure used across the studies. When confidence intervals were reported as the sole measure of variation for an effect size, they were transformed to PPP$, and in turn, were converted to standard errors. When only standard errors were reported in the original text, we kept the reported standard errors as the best approximation of variability. For those studies reporting a median estimate, the mean MWTP estimates were calculated from the interquartile range. When MWTP was expressed as a percentage [30,31,32,33], we used country-specific indexes and calculated an approximation of MWTP based on the products reported in each study. For those studies using consumer segmentation techniques based on variables that were relevant to the research questions (e.g., organic versus non-organic users, consumers segmented based on demographic characteristics or knowledge about ecolabels), the different consumer groups were combined, where this was possible, in order to obtain a pairwise comparison between intervention and control conditions. When the necessary information was not available, all eligible subgroups were imported into the analysis. In one study [34], the segmentation was not based on consumers’ characteristics (the origin of the product was used). In this study, the different subgroups were meta-analysed using a fixed-effects model in order to obtain two pairwise effect sizes (i.e., one for the control and one for the intervention arm regardless of the origin of the product). 

Data analysis was performed using Stata Version 16 [35]. Mean differences of MWTP (ecolabel vs. non-ecolabel) were computed for all of the included studies. A random-effects model (DerSimonian–Laird method) was adopted to capture study-specific effects [36]. We conducted a post-hoc subgroup analysis for each of our pre-specified subgroups; meat-dairy, seafood, fruits-vegetables-nuts. A meta-regression was pre-planned and conducted. The dependent variable was the reported MWTP, in PPP$ per kilogram, for food/milk products. Predictor variables included the setting within which each study was conducted (i.e., online or in-person), intervention attributes (i.e., message, format, and certification), and demographic characteristics (i.e., age, gender, income, education). The setting where the DCEs were conducted was coded as a dichotomous variable (0 = online, 1 = in-person), the type of product was treated as a categorical variable (1 = meat and dairy, 2 = seafood/fish, 3 = vegetable, fruits, and nuts), ecolabel certifications were coded as a dichotomous variable (0 = no certified, 1 = certified), message type was coded as a categorical variable (1 = organic, 2 = environmental sustainable, 3 = both), and the format of the label was coded as a categorical variable (1 = text, 2 = logo, 3 = both). All the demographic predictors were treated as continuous variables (percentages within pre-specified ranges, please see Appendix A).

## 3. Results

The search yielded 2904 abstracts following the removal of duplicates. Screening of title and abstracts resulted in 122 full-text articles for eligibility assessment, of which 43 were included in the review and 35 in the meta-analysis (Figure 1). Studies were excluded if they did not employ “willingness-to-pay” as an outcome measure or used a study design other than a discrete choice experiment.

### 3.1. Description of Studies

In total, 43 published papers reported 54 DCEs, representing data from 41,777 participants (Appendix A). Included in the meta-analysis were 35 papers reporting results from 35,725 participants. Five studies reported results from DCEs conducted in different countries. Ten papers reported studies conducted in North America, one in Central America, five in Japan, seven in China, one in Vietnam, one in Australia, five in the UK, and eighteen in Europe (see Appendix A for specific study locations).

The reference groups for demographic characteristics are: (i) percentage of consumers aged less than 40 with most of them aged up to 35, (ii) percentage of females, (iii) percentage of consumers with a household income at the lowest quartile according to the county where each DCE was conducted, and (iv) percentage of consumers with an undergraduate education or higher. 

### 3.2. Characteristics of Studies Included in the Meta-Analysis

Of the thirty-five papers included in the meta-analysis, twenty reported studies were conducted in an online environment. In-person interviews and/or hard copy questionnaires in virtual environments or in front of stores/supermarkets were used to collect data in fifteen papers. Ten papers reported studies that tested the effects of ecolabels on meat and dairy products, eleven papers reported studies on seafood and aquaculture products, and fifteen papers reported studies on vegetables, fruits, or nuts (Appendix A).

In sixteen papers, intervention conditions included ecolabels conveying an organic message only. Ten studies tested the effects of displaying values for specific environmental impact indicators, including CO_2_ emissions, water, land, pesticide use, and biodiversity loss. Nine studies tested the effects of both organic and environmental sustainability messages. Twenty-one out of thirty=five studies displayed a specific certification label, most commonly the USDA organic certification or MSC sustainable fisheries label. Ten studies tested an ecolabel combining different schemes (see Appendix A). 

The majority of participants were female (approximately 60%); a third (35%) were under 35 years old; 48% were highly educated (had attended tertiary education); and 25% of participants reported a household income equal to or lower than the lowest national quartile. These percentages were similar between the participants included in the meta-analysis and those included in the qualitative synthesis only (Appendix A).

### 3.3. Quality Assessment

All studies scored well against the International Society for Pharmacoeconomics and Outcomes Research (ISPOR) conjoint analysis checklist. The median score was 25 out of 30 and the range was from 21 to 28. Missing information was observed in the following categories: (i) elicitation format (category 5.2), (ii) respondents’ characteristics (category 8.1), and (iii) quality of responses (category 8.3).

### 3.4. Meta-Analysis 

In the primary analysis, thirty-five studies provided one-hundred-and-twenty-nine mean MWTP estimations. The mean MWTP for ecolabelled foods was 3.79 PPP$/kg, 95%CI = [2.7, 4.89], *p* ≤ 0.001, *I*^2^ = 100%, Tau^2^ = 39.46, *df* = 128, and *p* < 0.0001. Heterogeneity in the effects remained high even after excluding studies with the smallest effect sizes (*I*^2^ = 100%, Tau^2^ = 40.02, *df* = 113, and *p* < 0.0001). This heterogeneity was driven by magnitude rather than the direction of effect and will likely include geographic differences between studies and differences in the base price of products. Of the one-hundred-and-twenty-nine effect estimates, only five were in the direction indicative of a lower WTP for ecolabelled products. 

As serving size varies substantially between food groups we also conducted post hoc subgroup analyses to illustrate our main outcome (PPP$/kg) within each of our three pre-specified food categories. The MWTP for ecolabelled meat/dairy products was 9.24 PPP$/kg (95%CI: 7.21, 11.28); for seafood 2.71 PPP$/kg (95%CI: 2.3, 3.13); and for fruits, vegetables, and nuts 0.72 PPP$/kg (95%CI: 0.62, 0.82). Heterogeneity remained high (>99.5%) within each subgroup. 

### 3.5. Meta-Regression

Two pre-planned meta-regression models were conducted, testing the effects of intervention characteristics and demographic variables on the observed mean difference, and controlling for product type (see Table 1). 

Meta-regression models 1 and 2 have tested the effects of intervention and demographic characteristics, respectively, controlling for the product type.

All intervention characteristics tested, except certification labels and label format, were associated with differences in MWTP for ecolabelled foods and milk. The MWTP was also associated with the type of environmental sustainability messages. Sustainability labels were associated with lower MWTP compared to organic labels. Gender and education were associated with participants’ MWTP (Table 1). Women were willing to pay more than men for food/milk products with an ecolabel. Higher education was associated with lower MWTP for labelled food/milk products. Neither age nor income were associated with MWTP. Heterogeneity remained high across the different meta-regression models (Table 1).

### 3.6. Studies Excluded from the Quantitative Synthesis

Some studies did not present data in a way that could be included in the meta-analysis and these were considered separately. In line with the results of the main analysis, participants in all of these studies were willing to pay more for food and drinks with ecolabels (see Appendix A). In five of these studies, a known certification scheme was used [15,16,18,20,22] while three studies tested both organic and environmental sustainability labels [20,21,22]. In one study, negatively framed environmental sustainability messages resulted in lower MWTP [19].

## 4. Discussion

This meta-analysis showed that participants report a willingness to pay more for foods with an ecolabel. When measured as PPP$/kg, this effect was stronger for meat and dairy products compared to seafood, nuts, vegetables, and fruits. Organic labels were valued more highly than more specific environmental sustainability labels. Female participants and those from lower educational backgrounds were willing to pay a greater price premium for foods with an ecolabel. 

A number of methodological, intervention, and demographic variables were associated with participants’ WTP. We observed a higher MWTP for meat and dairy products compared to other food types which is consistent with a recent meta-regression of WTP for local food which found higher WTP for animal products and processed products compared to produce [9]. While food production, in general, can have negative health, social, and environmental implications, it may be that these issues are seen to be more relevant to consumers for meat than any other foods. Animal welfare, environmental impact, food safety, and health warnings have all been, and continue to be, common themes in public debates about meat, and less so about foods such as seafood or produce. Consumers may, therefore, be more open to paying a premium for meat that claims to eliminate these concerns to reduce the cognitive dissonance associated with meat consumption. However, since meat is typically consumed in relatively small quantities compared to other food groups, it is unclear whether this is an artefact of the measure we used (PPP$/g) and may not be present if using a different measure, e.g., PPP$/kcal. There were insufficient data in this study to investigate this. 

Products with an organic certification label were associated with higher MWTP compared to products conveying other types of messages; consistent with results from our companion review of selection, purchase, and consumption outcomes [10]. Organic certification labels have a much longer history compared to other labels included in this analysis, dating back to the 1980s in some countries. This likely increases consumer familiarity and trust with these labels as opposed to other claims. They may also signify wider attributes associated with product quality which speak to consumer values other than environmental sustainability that were not measured here. 

A combined ecolabel format (text and logo) was associated with higher MWTP compared to single-format ecolabels. However, these effects should be interpreted cautiously, given the inconsistent effect sizes in the meta-regressions and the fact they are derived from indirect comparisons.

Meta-regressions showed significant associations between participant characteristics and WTP for foods with an ecolabel. In line with a recent study [37], female and younger participants were found to be more responsive to ecolabels compared to male and older participants, respectively. In our review of selection, purchase and consumption outcomes, women also seemed more receptive to ecolabels, but results regarding age were mixed [10]. 

Other research has shown that women have a greater level of environmental concern and are more likely to adopt pro-environmental behaviours compared to men. In the current review, consumers aged up to 40 years were found to be more responsive to ecolabels; however, these findings should be interpreted cautiously as only 39 out of 129 effect sizes contributed to the analysis. There was evidence that people with lower levels of education exhibited higher WTP than those with higher education. However, the experimental nature of all of the studies included here means that the sample is biased towards participants with higher levels of education. In addition, the hypothetical scenarios may not be fully representative of true purchasing behaviour. 

To the best of our knowledge, this is the first systematic review to quantitatively synthesise the evidence from discrete choice experiments regarding the effects of ecolabels on participants’ MWTP. This review has several strengths. It was based on a pre-registered protocol and conducted in accordance with Cochrane methods. It also provides WTP estimations in an equalised purchase power unit (base year: 2018), facilitating the interpretation of the results across countries, populations, and conditions. 

### 4.1. Limitations

However, there are some limitations. First, we limited the search to discrete choice experiments and comparable approaches (such as best–worst scaling) as these are the most common experimental methods for eliciting WTP. WTP valuations derived from other approaches (such as experimental auctions) are not comparable due to differences in study design and were excluded. However, even endeavouring to synthesise these broadly comparable studies, the unexplained heterogeneity was high. This limits the generalisability and certainty of our findings, though the direction of effect was remarkably consistent across comparisons. Second, only databases of studies published in English were searched at follow-up. Finally, we used data solely derived from DCEs based on stated preference data. This study design is thought to yield upward-biased estimates of WTP^14^. As such, our estimates might be better thought of as an upper limit on WTP. However, this is less likely to affect the relative importance of features in the subgroup analysis and meta-regressions. 

### 4.2. Conclusions

These results show that consumers are willing to pay more for environmentally sustainable products. In combination with our previous review [10], which showed that ecolabels can increase the selection and purchase of food products, these comprehensive reviews of the corpus of evidence suggest ecolabels are a promising strategy to encourage more sustainable purchasing. However, most of the research to date is based on experimental studies in virtual settings. Further research is needed directly comparing different formats and types of ecolabels in real-world settings to provide more robust evidence of effectiveness.

## Figures and Tables

**Figure 1 nutrients-13-02677-f001:**
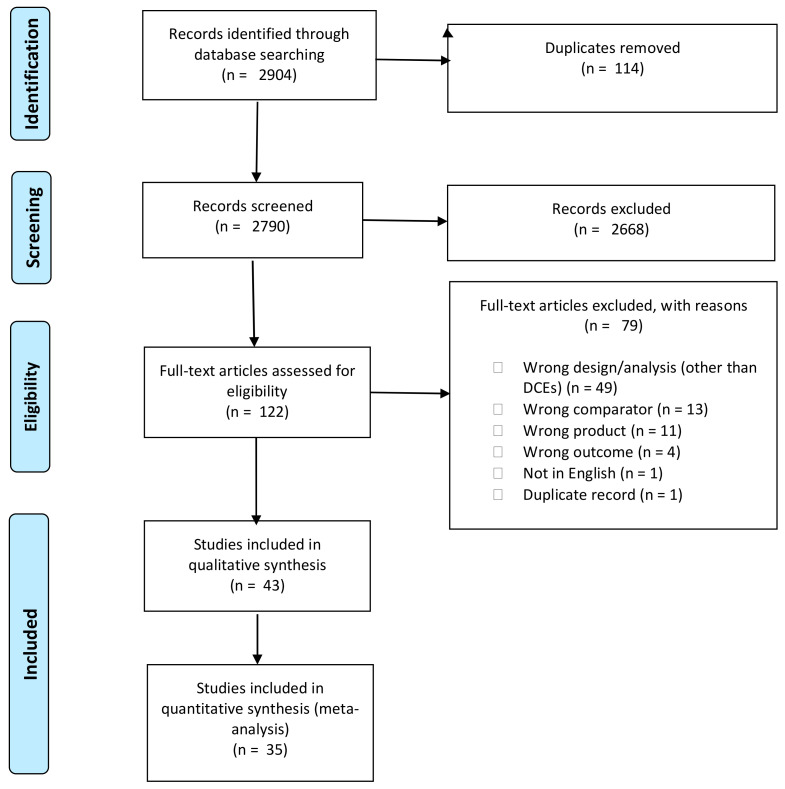
PRISMA flow diagram.

**Table 1 nutrients-13-02677-t001:** Meta-regression results (measuring MWTP in PPP$/kg).

	Meta-Regression 1 Interverntions’ Characteristics Coefficients (95%CI)	Meta-Regression 2 Participants’ Characteristics Coefficients (95%CI)
Setting(in-person interview/questionnaire)	3.48 * (0.14, 6.82)	
Message(reference group: Organic)		
Sustainability	−3.62 * (−6.79, −0.44)	
Combined	−2.60 (−6.52, 1.32)	
Certification(reference group: certification label present)	2 (−1.19, 5.21)	
Format (reference group: text format)		
Label	−0.72 (−4.65, 3.2)	
Combined	−0.84 (−4.15, 2.46)	
Age		6.6 (−13.48, 26.7)
Gender: Female		28.25 *** (12.83, 43.67)
Income		3.25 (−14.29, 20.81)
Education		−28.81 *** (−36.95, −20.67)
Obs.	129	39
R^2^ (%)	7.45	56.41
T^2^	53.19	11.27
I^2^ (%)	100	100

Note: Dependent variable is M WTP for labelled food and milk. *, *** indicate statistical at 0.05 and 0.001, respectively.

## Data Availability

All data (reviewed articles) used, were from already published empirical articles, retrieved from databases. All data generated or analysed during this study were included in this published article.

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
