# Peer review of "The Impact of Environmental Sustainability Labels on Willingness-to-Pay for Foods: A Systematic Review and Meta-Analysis of Discrete Choice Experiments"

_nutrients, 2021, doi:10.3390/nu13082677_

Round 1

Reviewer 1 Report

First of all, I would like to congratulate to the authors for present a good paper. 
The aim of this manuscript is line with the methods and the results presented. The main aim is to analyses the empirical evidence regarding the effect of price on products bearing ecolabels. The procedure conducted on the systematic-review and the meta-analysis is one of the core strengths of this work.
However, to increase the effectiveness of this manuscript I suggest to authors, extend the introduction section developing the idea of ecolabel as instrument of sustainability and public health policy.
Material and Methods. Please check the register code, in PROSPERO web, this code corresponds to other protocol.
In line 78, authors indicated that considered two settings: stores/supermarkets and virtual environments. However, in line 206, authors said mention laboratories setting. Please, check this incongruence.
Authors conducted two meta-regressions, one with intervention's characteristics and other one with participants' characteristics. Why authors did not estimate one meta-regression with all variables? Could this separation of the variables cause part of the high level of heterogeneity? Please include the criteria considered to decide this separation.
I think that standardize the wtp as PPP$ is a right decision. However, I wonder if the Big Mac Index is better rather than a generic PPP$.

Author Response

Reviewer 1 Comments

Response

First of all, I would like to congratulate to the authors for present a good paper. 
The aim of this manuscript is line with the methods and the results presented. The main aim is to analyses the empirical evidence regarding the effect of price on products bearing ecolabels. The procedure conducted on the systematic-review and the meta-analysis is one of the core strengths of this work.
However, to increase the effectiveness of this manuscript I suggest to authors, extend the introduction section developing the idea of ecolabel as instrument of sustainability and public health policy.

Thank you. We have added further text in the introduction positioning ecolabels from this perspective.

Material and Methods. Please check the register code, in PROSPERO web, this code corresponds to other protocol.

Apologies, we’ve updated the manuscript with the correct Prospero code (CRD420180943300).

In line 78, authors indicated that considered two settings: stores/supermarkets and virtual environments. However, in line 206, authors said mention laboratories setting. Please, check this incongruence.

We have replaced “laboratories” with “virtual environments” to make this consistent.

Authors conducted two meta-regressions, one with intervention's characteristics and other one with participants' characteristics. Why authors did not estimate one meta-regression with all variables? Could this separation of the variables cause part of the high level of heterogeneity? Please include the criteria considered to decide this separation.

We conducted a meta-regression with all variables included in the model, however the heterogeneity was still substantial. Also, a meta-regression with all variables included in the model was less powered given that studies with even a missing variable were excluded from the analysis.

Our strategy was to disentangle the effects of products’ properties from demographic factors which we think was wise, taking into account the large heterogeneity of countries and product categories included in the review. Perhaps it would have made more sense to provide a meta-regression model with all variables if we were assessing the effects of a specific product type (meat) in a specific country.

I think that standardize the wtp as PPP$ is a right decision. However, I wonder if the Big Mac Index is better rather than a generic PPP$.

This is a fair point. Our preference for PPP$ is that it is viewed as a formal tool, backed by a history of use in analyses by the OECD. Their database also provided us with a sufficient depth of detail to accommodate the wide range of countries from which our studies were drawn.  

Reviewer 2 Report

The authors in the materials and methods section literally say “We conducted a systematic review and meta-analysis of discrete choice experi-60 ments. Reporting follows the Preferred Reporting Items for Systematic Reviews and 61 Meta-Analyses (PRISMA) checklist”.

Well, the PRISMA statement, in the 2009 version (ítem 8), clearly says “Present full electronic search strategy for at least one database, including any limits used, such that it could be repeated” and in the 2020 declaration (ítem 7) he insists on this point but more strictly “Present the full search strategies for all databases, registers and websites, including any filters and limits used”. But, in the present review neither of the two possibilities has been fulfilled, which invalidates the method or at least does not fulfill what the authors say.

It is not specified whether the selection of the studies was made by one auotor or by several and, in the latter case, the relationship between their selections.

The same can be said of the study of the quality of the selected articles.

In the results section, in the flow chart, it is mentioned that 2669 records have been eliminated, but the reason for these exclusions is not fully detailed.

These methodological flaws are sufficient for this review not to be fully reproducible.

Author Response

Reviewer 2 Comments

Response

The authors in the materials and methods section literally say “We conducted a systematic review and meta-analysis of discrete choice experi-60 ments. Reporting follows the Preferred Reporting Items for Systematic Reviews and 61 Meta-Analyses (PRISMA) checklist”.

Well, the PRISMA statement, in the 2009 version (ítem 8), clearly says “Present full electronic search strategy for at least one database, including any limits used, such that it could be repeated” and in the 2020 declaration (ítem 7) he insists on this point but more strictly “Present the full search strategies for all databases, registers and websites, including any filters and limits used”. But, in the present review neither of the two possibilities has been fulfilled, which invalidates the method or at least does not fulfill what the authors say.

Our Prospero protocol (CRD42018094330) provides details on the search strategy, including keywords, limits and databases used and is referenced in the methods section. We were trying to avoid duplication, but we are very happy to repeat the details of the search strategy etc within the paper if the Editor wishes us to do so.

It is not specified whether the selection of the studies was made by one auotor or by several and, in the latter case, the relationship between their selections.

The selection of studies was made independently by two study authors and any disagreement was resolved with discussion and/or referral to a third reviewer. We reached an agreement after discussion in all cases, so there was no reason to ask advice from a third reviewer. Section 2.2 “Data extraction & quality assessment” includes information on our process.

The same can be said of the study of the quality of the selected articles.

Two authors rated quality using the ISPOR checklist. We’ve added this for clarity to section 2.2 “2.2 Data extraction & quality assessment”.

In the results section, in the flow chart, it is mentioned that 2669 records have been eliminated, but the reason for these exclusions is not fully detailed.

We have added in further details about reasons for exclusions into section 3 “Results”.

These methodological flaws are sufficient for this review not to be fully reproducible.

We believe that the existing manuscript along with the amendments described above should be sufficient for future researchers to reproduce our searches and screening.  We are very willing to add any further information requested by the Editors.

Reviewer 3 Report

Dear Authors,

The manuscript (nutrients-1268809) presented for review is interesting and the topic of the manuscript is very up to date. I recommend the article for publication after major revision by the Authors.

Authors, Please note and address the following comments:

Introduction

The topic of the article is interesting. However, the manuscript should be improved in section Introduction. In my opinion, the introduction should be a strong connection with the manuscript. The Introduction did not show a gap in the available results in the literature.

Materials and methods

What is the reason of excluded 2669 records? Despite this chapter is described in detail, the authors did not write about why excluded these records.

Results

The technical quality of the article is low. Figure 1, and Table 2 in this form are not clear (small fonts and blurred). Therefore the manuscript becomes incomprehensible to readers.

In my opinion, the last column in the Table titled 'Study characteristics' presented in Appendix A  is not completely understandable, too. The authors should be improved this.

Limitation (lines 329-340)

The authors wrote about the limitations of their meta-analysis in lines 329-340. Other authors usually prepare separate section Limitations. I reckon that is a good idea.

Conclusion

In this manuscript, there is a lack of section Conclusion. In my opinion, the text in lines 342-346 is looking good for this chapter. It seems that a clear separation of chapter Conclusions would be well. Moreover, conclusions should be the response to the aims presented in lines 53-58.

What are the practical and theoretical implications of this analysis? What should be the directions of further research in this range?

 References

Authors, Please check the correctness of the citation of references by requirements of the Nutrients Journal. Now, the citation in the text is not appropriate, the authors used the top index instead of a number in square brackets. Moreover, in the section References - should be used publication year after the name of the journal. Now the publication year is after the names of authors.

Despite my comments, I am pleased to recommend this manuscript for publication. The article needs to be improved. I hope that these comments will help the authors with their work under improving the manuscript.

Reviewer

Author Response

Reviewer 3 Comments

Response

Dear Authors,

The manuscript (nutrients-1268809) presented for review is interesting and the topic of the manuscript is very up to date. I recommend the article for publication after major revision by the Authors.

Authors, Please note and address the following comments:

Introduction

The topic of the article is interesting. However, the manuscript should be improved in section Introduction. In my opinion, the introduction should be a strong connection with the manuscript. The Introduction did not show a gap in the available results in the literature.

Our scoping of the literature before undertaking the project found no reviews of WTP for these types of products. We feel that we have commented on the gaps in the literature in the introduction:

“Though some studies have tested consumers’ willingness-to-pay (WTP) for ecolabelled products, there is a lack of consolidated data on how exposure to different types of ecolabels affects the perceived value of a product, expressed in monetary terms, and whether this varies with demographic variables such as age, gender, or income, or for different types of products.”

Materials and methods

What is the reason of excluded 2669 records? Despite this chapter is described in detail, the authors did not write about why excluded these records.

We have added further text in the section 3 (Results) that explains reasons for excluding these papers.

“Studies were excluded if they did not employ “willingness-to-pay” as an outcome measure or used a study design other than a discrete choice experiment.”

Results

The technical quality of the article is low. Figure 1, and Table 2 in this form are not clear (small fonts and blurred). Therefore the manuscript becomes incomprehensible to readers.

In my opinion, the last column in the Table titled 'Study characteristics' presented in Appendix A  is not completely understandable, too. The authors should be improved this.

Thank you for the feedback. We have increased the sizes of Figure 1 and Table 2 to the maximum that the formatting allows.

Limitation (lines 329-340)

The authors wrote about the limitations of their meta-analysis in lines 329-340. Other authors usually prepare separate section Limitations. I reckon that is a good idea.

We have included a separate limitations section as 4.1.

Conclusion

In this manuscript, there is a lack of section Conclusion. In my opinion, the text in lines 342-346 is looking good for this chapter. It seems that a clear separation of chapter Conclusions would be well. Moreover, conclusions should be the response to the aims presented in lines 53-58.

What are the practical and theoretical implications of this analysis? What should be the directions of further research in this range?

References

Authors, Please check the correctness of the citation of references by requirements of the Nutrients Journal. Now, the citation in the text is not appropriate, the authors used the top index instead of a number in square brackets. Moreover, in the section References - should be used publication year after the name of the journal. Now the publication year is after the names of authors.

Despite my comments, I am pleased to recommend this manuscript for publication. The article needs to be improved. I hope that these comments will help the authors with their work under improving the manuscript.

Thank you for flagging this. The citation formats have been updated.

Reviewer 4 Report

The article “The Impact of Environmental Sustainability Labels on Willingness-To-Pay for Foods: A Systematic Review and Meta-Analysis of Discrete Choice Experiments” is well structured in terms of the parts required of articles of this nature. The part dedicated to the introduction is too brief. It would be useful to include more bibliographic references about the labelling of organic products and how this compares to other types of labelling that bestow quality. Little attention is paid to the concept of organic food with labelling and the social values associated with this type of consumer behaviour. To correct this weakness, we recommend including the reference:

Muñoz-Sánchez, V.-M.; Pérez-Flores, A.-M. The Connections between Ecological Values and Organic Food: Bibliometric Analysis and Systematic Review at the Start of the 21st Century. Sustainability 2021, 13, 3616. https://doi.org/10.3390/su13073616.

Turning to the methodology section, the article makes good use of the material and databases. The use of PRISMA is appropriate and the analysis method applied to the results is suitable. The databases consulted are extensive and cover the main areas of exploration at the international level. The choice of subject, objectives and the proposed hypotheses are all appropriate, and the way the article is developed is pertinent. The statistical regression analysis is applied adequately with the data being interpreted correctly. In my opinion, the figures in table 2 are too small, making reading difficult: however, this may be due to formatting instructions.

One positive in the section dedicated to discussion is that it highlights the most contentious aspects of the research. It would, perhaps, be useful to refer briefly to the weaknesses in the methodological design or the scope of the results. The references section is well prepared, being both sufficiently up-to-date and extensive. However, some references are preceded by an asterisk, showing that they belong to the group of references used in the meta-analysis, when there is no need for them to be highlighted using this symbol.

Therefore, we recommend publication of the paper with the small changes suggested to sharpen its existing quality. 

Author Response

Reviewer 4 Comments

Response

The article “The Impact of Environmental Sustainability Labels on Willingness-To-Pay for Foods: A Systematic Review and Meta-Analysis of Discrete Choice Experiments” is well structured in terms of the parts required of articles of this nature. The part dedicated to the introduction is too brief. It would be useful to include more bibliographic references about the labelling of organic products and how this compares to other types of labelling that bestow quality. Little attention is paid to the concept of organic food with labelling and the social values associated with this type of consumer behaviour. To correct this weakness, we recommend including the reference:

Muñoz-Sánchez, V.-M.; Pérez-Flores, A.-M. The Connections between Ecological Values and Organic Food: Bibliometric Analysis and Systematic Review at the Start of the 21st Century. Sustainability 2021, 13, 3616. https://doi.org/10.3390/su13073616.

Thank you, we have added a bit more context in the introduction about the role of ecolabels. Unfortunately, the suggested review paper has more of a bibliometric focus so does not provide the suggested content that we feel is appropriate to add to the manuscript.

Turning to the methodology section, the article makes good use of the material and databases. The use of PRISMA is appropriate and the analysis method applied to the results is suitable. The databases consulted are extensive and cover the main areas of exploration at the international level. The choice of subject, objectives and the proposed hypotheses are all appropriate, and the way the article is developed is pertinent. The statistical regression analysis is applied adequately with the data being interpreted correctly. In my opinion, the figures in table 2 are too small, making reading difficult: however, this may be due to formatting instructions.

One positive in the section dedicated to discussion is that it highlights the most contentious aspects of the research. It would, perhaps, be useful to refer briefly to the weaknesses in the methodological design or the scope of the results. The references section is well prepared, being both sufficiently up-to-date and extensive. However, some references are preceded by an asterisk, showing that they belong to the group of references used in the meta-analysis, when there is no need for them to be highlighted using this symbol.

Therefore, we recommend publication of the paper with the small changes suggested to sharpen its existing quality. 

Thank you for the comments. We have increased the size of Table 2 as much as the formatting allows, hopefully this makes the table’s content more accessible for readers.

Round 2

Reviewer 2 Report

The methodology has not been improved as requested. The work remains non-reproducible by other authors or readers.

Author Response

To aid reproducibility, we have added Appendix B with detailed information on searches in four databases.

In Section 2.1 we added further details about the search strategy:

An updated, more precise, follow-up search adding in search filters specific to studies measuring willingness-to-pay17 was conducted on 8 November 2019 on the same databases. The search filters consisted of the following terms: ‘discrete choice experiment”, ‘discrete choice experiments’, ‘discrete choice modelling’, ‘discrete choice modelling’, ‘discrete choice conjoint experiment’, 'discrete-choice experiment', ‘discrete-choice experiments’, conjoint analysis’, ‘conjoint measurement’, ‘conjoint studies’, ‘conjoint choice experiment’, ‘conjoint choice experiments’, Best-Worst Scaling’, ‘Best Worst Scaling’, ‘MaxDiff Scaling’, ‘Maximum Difference Scaling’, ‘Contingent valuation’.

In Section 2.2. we added further details about the data extraction process:

Two independent reviewers (AB & SK/CP/FB) conducted full-text screening with a high-level of agreement (κ =0.96). Two reviewers extracted data independently and disagreements were resolved by discussion (level of agreement in extraction phase > 85%).

Reviewer 3 Report

Dear Authors,

The authors have changed many parts: Introduction, Material, and methods, Conclusion, Results, Limitation, References of the planned paper in line with my and other reviewer's suggestions. In my opinion, now the manuscript is clear and better understand than the previous version, but still, Figure 1 and Table 2 aren’t clear. Please upload tables as table format, not as figure format.

I would like to thank the authors for considering my comments to improve their manuscript.

Author Response

Thank you. We have recreated Figure 1 and Table 2 with versions that we hope are much higher quality now.